# Probabilistic Tsunami Hazard Analysis for Vancouver Island Coast Using Stochastic Rupture Models for the Cascadia Subduction Earthquakes

Katsuichiro Goda [1,2] 

1 Department of Earth Sciences, Western University, London, ON N6A 3K7, Canada; kgoda2@uwo.ca; Tel.: +1-519-661-2111 (ext. 83189)
2 Department of Statistical and Actuarial Sciences, Western University, London, ON N6A 3K7, Canada

**Abstract:** Tsunami hazard analysis is an essential step for designing buildings and infrastructure and for safeguarding people and assets in coastal areas. Coastal communities on Vancouver Island are under threat from the Cascadia megathrust earthquakes and tsunamis. Due to the deterministic nature of current megathrust earthquake scenarios, probabilistic tsunami hazard analysis has not been conducted for the coast of Vancouver Island. To address this research gap, this study presents a new probabilistic tsunami hazard model for Vancouver Island from the Cascadia megathrust subduction events. To account for uncertainties of the possible rupture scenarios more comprehensively, time-dependent earthquake occurrence modeling and stochastic rupture modeling are integrated. The time-dependent earthquake model can capture a multi-modal distribution of inter-arrival time data on the Cascadia megathrust events. On the other hand, the stochastic rupture model can consider variable fault geometry, position, and earthquake slip distribution within the subduction zone. The results indicate that the consideration of different inter-arrival time distributions can result in noticeable differences in terms of site-specific tsunami hazard curves and uniform tsunami hazard curves at different return period levels. At present, the use of the one-component renewal model tends to overestimate the tsunami hazard values compared to the three-component Gaussian mixture model. With the increase in the elapsed time since the last event and the duration of tsunami hazard assessment, the differences tend to be smaller. Inspecting the regional variability of the tsunami hazards, specific segments of the Vancouver Island coast are likely to experience higher tsunami hazards due to the directed tsunami waves from the main subduction zone and due to the local underwater topography.

**Keywords:** probabilistic tsunami hazard analysis; stochastic rupture models; time-dependent earthquake occurrence; Cascadia subduction zone; Vancouver Island

## 1. Introduction

In the Pacific Northwest, the Cascadia subduction zone is widely recognized as a high-potential source for megathrust subduction earthquakes and tsunamis [1]. The driving mechanism is the subduction of the Juan de Fuca, Gorda, and Explorer Plates underneath the North American Plate, and the subduction zone spans from Vancouver Island to northern California. The most recent historical event occurred in January 1700, with an estimated moment magnitude (*M*) of 9 [2]. Since the identification of this potential source, various geological and geophysical pieces of evidence have been collected to characterize the recurrence and rupture patterns of the Cascadia megathrust events [3–6]. Coastal communities in the Pacific Northwest require reliable earthquake and tsunami hazard assessments to enhance disaster preparedness and resilience [7].

Focusing upon Vancouver Island of British Columbia, Canada, several studies have been conducted to quantify the potential tsunami hazards due to the Cascadia megathrust events [8–11]. A recent study by [12] considered three rupture types of the Cascadia

megathrust events, i.e., buried rupture, splay-fault rupture, and trench-breaching rupture, to evaluate the tsunami hazards for coastal locations around Vancouver Island. On the other hand, five tsunami scenarios that were developed by [13], having earthquake magnitudes between 8.7 and 9.3, were considered by [14] to evaluate tsunami hazards. The above-mentioned studies for Vancouver Island are scenario-based (deterministic) tsunami hazard assessments, lacking the consideration of a variety of possible earthquake rupture scenarios and their corresponding occurrence probabilities. To address this limitation, a comprehensive set of 5000 stochastic rupture models having the moment magnitudes between 8.1 and 9.1 (i.e., 500 models per 0.1 magnitude bin) was developed by [15], considering different rupture patterns for the Cascadia subduction earthquakes.

The next logical step for quantifying regional tsunami hazards on Vancouver Island is to conduct probabilistic tsunami hazard analysis (PTHA), which typically requires an earthquake occurrence model, earthquake rupture model, tsunami propagation model, and logic tree model for epistemic uncertain models and parameters [16,17]. To date, PTHA investigations for Vancouver Island have been hindered by the scenario-based nature of the existing tsunami hazard studies. The deficiency is that earthquake occurrence modeling of the Cascadia megathrust events has not been integrated with scenario-based tsunami hazard assessments. Thereby, the probabilities of earthquake occurrence are not assigned to these scenarios directly. To characterize earthquake occurrence of the Cascadia megathrust events, earthquake histories can be constructed from onshore subsidence records [3] and offshore marine turbidite records [6]. On average, full rupture of the Cascadia subduction zone is expected to occur every 530 years (19 events over 10,000 years), although notable clustering patterns and seismic gaps have been recognized in the geological history data [6]. Therefore, a time-dependent earthquake occurrence model, such as the Brownian Passage Time distribution [18] and Weibull distribution [19], may be more suitable for the Cascadia megathrust events. Regarding the use of geological earthquake event data, proper treatment of uncertainty of these data is necessary as the event dates are often estimated through radiocarbon analyses of sediment samples and geological cores [20,21].

This study presents regional PTHA of Vancouver Island due to the Cascadia megathrust ruptures as a novel contribution to the literature. The PTHA approach is based on time-dependent earthquake occurrence modeling [22] and stochastic rupture modeling [15]. This is the first PTHA study for the Canadian coast of the Cascadia region with comprehensive consideration of numerous potential seismic tsunami sources and constitutes an important step towards fully probabilistic tsunami hazard and risk assessments for coastal communities on Vancouver Island. The earthquake occurrence model considers a standard renewal process by characterizing the inter-arrival time between successive events using the Weibull distribution in comparison with the time-independent Poisson model. Moreover, by accounting for the uncertainty of the geological data and inaccuracy of dating techniques, Monte Carlo resampling is implemented to determine the sampling distribution of the inter-arrival time for the Cascadia megathrust events [21]. Subsequently, a three-component Gaussian mixture distribution is used to characterize the resampled data. For the earthquake rupture model, stochastic source modeling is adopted. Considering the full-margin ruptures of the Cascadia subduction zone (which extends to the northern segment), 2000 stochastic rupture models having the moment magnitudes between 8.7 and 9.1 are employed, and tsunami propagation simulations are conducted at the regional scale. The focus on the full-margin rupture scenarios is justified because tsunamis caused by central and southern margins of the Cascadia subduction zone (but not extending to the northern segment) are unlikely to generate significant tsunamis along the Canadian coast due to their tsunami radiation patterns [15]. The stochastic rupture models account for variable fault rupture geometry (e.g., length, width, and position) and heterogenous earthquake slip distributions. By taking the maximum wave amplitude as a tsunami intensity measure, PTHA results are presented as tsunami hazard curves for specific locations and as uniform tsunami hazard profiles for nearshore locations along Vancouver Island. By considering several variations of the earthquake occurrence model (e.g., inter-arrival time

distribution and magnitude distribution), sensitivity analysis is carried out to identify the influential models and parameters on PTHA results and to discuss the implications of these model assumptions. Section 2 introduces the tectonic characteristics of the Cascadia subduction region and the geological data on the full-margin megathrust ruptures. Section 3 presents a probabilistic tsunami hazard model for the Cascadia megathrust ruptures and explains its components, including the earthquake occurrence model, stochastic tsunami simulations, and computational procedures. Section 4 discusses regional tsunami hazard assessment results for coastal locations along the Vancouver Island coast, followed by the main conclusions of this study (Section 5).

## 2. Cascadia Subduction Zone

### 2.1. Tectonic Characteristics

The Cascadia subduction zone is one of the major sources of seismic and tsunami hazards for the Pacific coast of British Columbia, Canada. The Cascadia zone involves the eastward subduction of the oceanic plates beneath the continental plate. The oceanic plates move northeast with respect to the continental plate with relative velocities of 30 to 40 mm/year [23] (Figure 1a). The active crustal process of the Cascadia region exhibits gradual inter-seismic uplift and sudden coseismic subsidence along the Northwest coast.

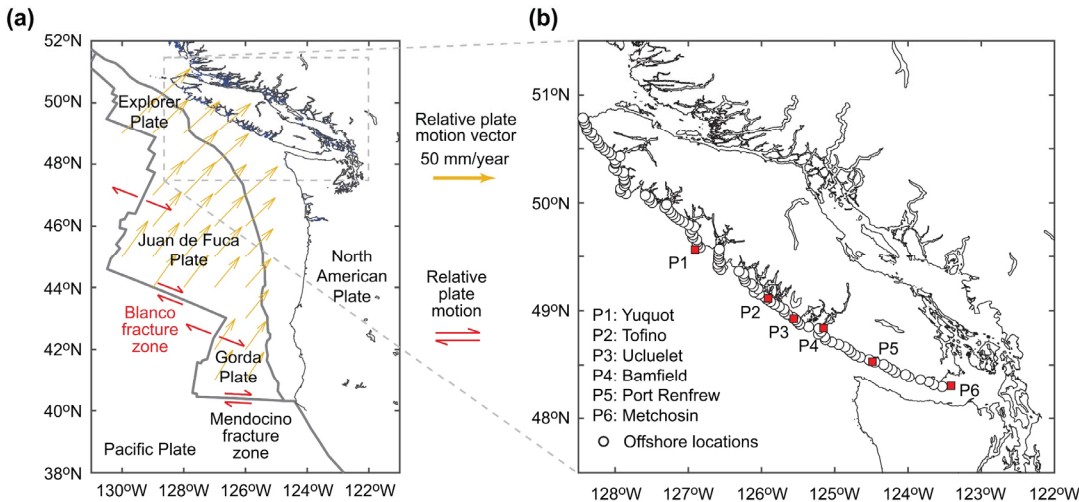

**Figure 1.** (**a**) Relative plate motion vectors of the Juan de Fuca, Gorda, and Explorer Plates with respect to the North American Plate in the Cascadia subduction zone. The relative plate motion between the oceanic and continental plates is based on [23]. (**b**) Locations of the wave recording sites along the Vancouver Island coast.

Evidence of earthquake-induced subsidence includes hollowed tree trunks in tidal marshes and tree rings exhibiting sudden death [3]. Collecting evidence of coseismic subsidence along the Cascadia coast is useful for constraining the spatial extent of the fault rupture and helps estimate the magnitudes of past megathrust earthquakes (i.e., rupture length and earthquake magnitude are correlated). Tsunami deposits preserved in environments that do not typically have considerable sedimental inflow can be used to estimate tsunami run-ups from past events.

Moreover, the earthquake slip of the rupture zone can be inferred from historical tsunami heights. Fault slip can be constrained using the coseismic vertical displacements observed in past events. The last significant earthquake along the Cascadia subduction zone occurred in 1700 with *M*9 [2,3], which ruptured the entire subduction margin. Evidence for the 1700 tsunami comes in the form of First Nation's myths in North America and written documents that recorded the damage and flooding in Japan. The sedimentary records of this event can be seen in the form of turbidity deposits, buried soils, and marsh data [3,6].

The ground subsidence observations for the 1700 event and the pre-1700 events are shown in Figure 2 [24].

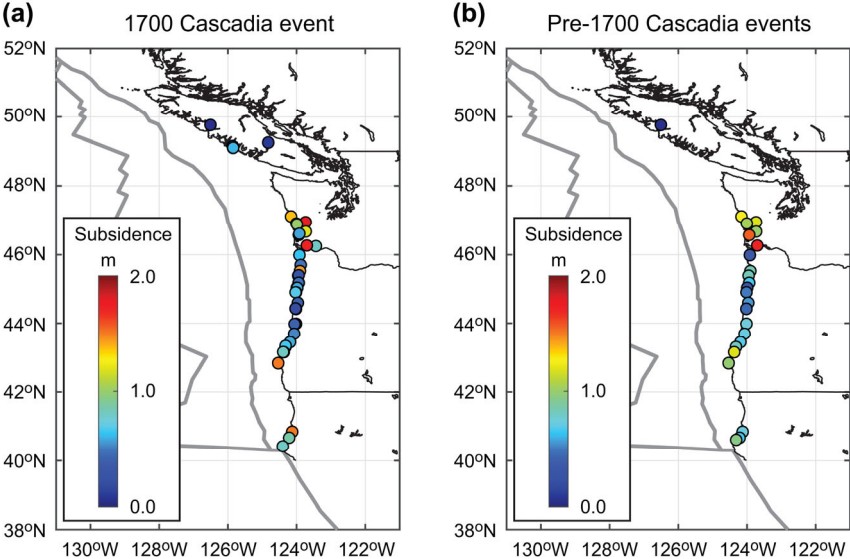

**Figure 2.** Ground subsidence observations based on various previous studies [24]: (**a**) 1700 event and (**b**) pre-1700 events.

## 2.2. Earthquake History Data Based on Offshore Turbidite Records

The Cascadia subduction region extends from Vancouver Island to Mendocino fracture zone (Figure 1a). Synchronously triggered turbidity deposits along the continental margin, where multiple submarine channel-canyon systems are distributed, provide a recurrence history of past megathrust events over the latest 10,000 years [6]. During major storms, earthquakes, and tsunamis, turbidity currents can be triggered by river-carried sandy and silty sediments sliding down the continental shelf. From branching tributaries, turbidity currents merge into the main channel and form a large turbidite. By contrast, small-to-moderate storms and far-field tsunamis are unlikely to induce synchronized turbidity currents along the entire continental margin of the Cascadia subduction zone.

To develop a catalog of megathrust subduction earthquakes in Cascadia, Goldfinger et al. [6] carried out extensive coring surveys spanning the entire margin of the Cascadia subduction zone and analyzed collected samples using marine radiocarbon dating and stratigraphic correlation techniques. Due to the synchronous occurrence of turbidite currents along the northern half of the Cascadia coast, the northern Cascadia events are best explained by paleo-seismic events, and corresponding paleo-seismic events can be found in the southern half of the Cascadia coast (i.e., these were full-margin Cascadia events). For the southern Cascadia subduction zone, most of the turbidite samples are well-corresponded and correlated with the spatial extent of shorter onshore paleo-seismic records, while there are uncorrelated turbidites that were likely to be depositional products after smaller earthquakes, local storms, or far-field tsunamis. The northern portion of the Cascadia subduction zone ruptured less frequently with recurrence periods of 500 to 530 years and has a strong spatial correlation with the southern half (determined based on the estimated radiocarbon dates of different geological cores along the subduction margin), thus leading to a synchronized full or near-full rupture which could result in *M*9-class megathrust events. The southern portion of the Cascadia subduction zone, in addition to the whole-region ruptures, experienced additional smaller earthquakes (*M*8-class events) according to the turbidite records with recurrence periods of 240 to 320 years. Nineteen well-dated turbidite events were identified and were thought to be triggered by the northern (i.e., full or near full margin) Cascadia ruptures (denoted by T1 to T18 plus T17a that was considered as a separate full-margin rupture from T17) [6]. Note that T1 corresponds to

the most recent 1700 event, whereas T18 corresponds to the oldest event in 9795 calibrated years before the present.

### 2.3. Resampled Turbidite-Based Earthquake History Data

Kulkarni et al. [21] adopted the full-margin rupture data by [6] to develop an earthquake clustering model for the Cascadia subduction zone, and the data are shown in Figure 3a. The data consist of 83 turbidite ages for the 19 events (i.e., T1 to T18, including T2 and T17a). Different events have different numbers of age data (for instance, T3 has eight age data). Each of the turbidite pieces of data comes with three age estimates, i.e., best, +2 sigma bound, and −2 sigma bound. The probability distribution of the individual age data can be represented by the triangular distribution with the best estimate as the mode and the two-sigma bounds as the upper and lower limits [21]. Kulkarni et al. [21] observed several gaps between subsequent full-rupture events and were motivated to develop an earthquake occurrence model that distinguishes inter-cluster data and gap data using a hierarchical clustering method. Among the 19 turbidite events, T2 was eventually excluded from the analysis because this event was not consistently recorded in buried soil/marsh data on land, implying that T2 might have been caused by non-seismic sources. After excluding T2 from the dataset, they obtained the 17 inter-arrival time data from the 18 age data.

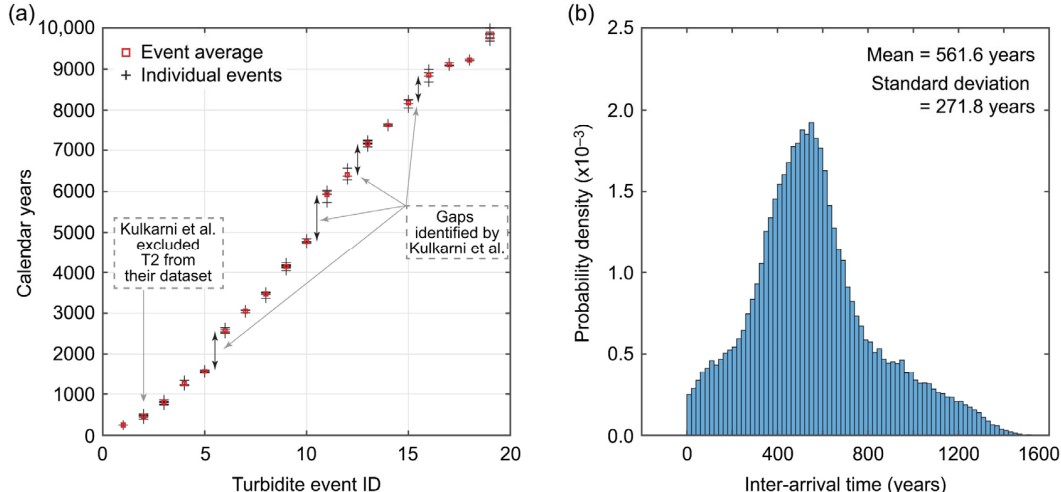

**Figure 3.** (**a**) Cascadia age data by [6]. (**b**) Histogram of the inter-arrival time data from 5000 resampled earthquake history data, following the same approach taken by [21]. The simulated earthquake catalogs that contain negative inter-arrival times are excluded.

Due to nonnegligible uncertainty of the Cascadia age data from [6], Kulkarni et al. [21] conducted 20 Monte Carlo resampling of the Cascadia age data and used it for their hierarchical clustering analysis. The procedure of the resampling is summarized as follows:

1.  Set the number of Monte Carlo resampling.
2.  For each turbidite event, choose one of the pieces of data randomly with equal chance (i.e., all listed data in [21] are regarded as equally reliable).
3.  Sample the age of the turbidite data chosen in Step 2 from the triangle distribution, which is determined by the best and +/− 2 sigma bounds.
4.  Repeat Steps 2 and 3 for all turbidite data (i.e., T1 to T18). The inter-arrival time data can be obtained for each catalog.
5.  Repeat Steps 2 to 4 for the resampling number specified in Step 1.

Because the radiocarbon-dated ages involve large uncertainty and some of the adjacent turbidite events are only separated by a few hundred years, the above-mentioned resampling procedure can result in the reversed order of the simulated events. When the simulated Cascadia age catalog has negative inter-arrival time data, the simulated trial is discarded. Note that in [21], simulated Cascadia age catalogs that have inter-arrival

time data longer than 100 years only were considered because the shortest inter-arrival time in the original Cascadia age catalog was about 100 years (Figure 3a). This resampling method results in a high rejection rate of the simulated catalogs (approximately 62% of the simulated catalogs are abandoned). Since the threshold of 100 years is arbitrary, and this high rejection rate could cause bias in statistical modeling, all positive inter-arrival time data are adopted instead of data being greater than 100 years. In this case, the rejection rate is approximately 20%. The considerations of a larger resampling size (=5000) and a less subjective rejection criterion of the simulated turbidite age data lead to robust characteristics of the simulated Cascadia event time data.

Figure 3b shows a histogram of the Cascadia inter-arrival time distribution from 5000 resampling simulations. Each simulated catalog consists of 18 events, and 17 inter-arrival time data can be calculated. The simulated inter-arrival time data exhibit heavy tails on both upper and lower ends, compared with the normal distribution. The longer inter-arrival time data are associated with long gaps, whereas the shorter inter-arrival time data are related to short-time clustering. Figure 3b shows the right skewness of the inter-arrival time data, which is influenced by the long gaps between clustered sets of events. The mean and standard deviation of the simulated inter-arrival time data are 561 years and 272 years, respectively, and thus the coefficient of variation is calculated as 0.485. The resampled inter-arrival time data can be used to determine the goodness-of-the-fit for different earthquake occurrence models (Section 3.1).

## 3. Probabilistic Tsunami Hazard Model for the Cascadia Subduction Earthquakes

Tsunami hazard analysis is an essential step for designing buildings and infrastructure and for safeguarding people and assets in coastal areas. It determines relevant tsunami intensity measures at a nearshore location that need to be considered for calculating tsunami loads acting on coastal structures. In the last two decades, probabilistic methods have been introduced in tsunami structural design codes and guidelines as alternatives to conventional deterministic methods. The necessity of probabilistic approaches was motivated by the catastrophic tsunami events in the Indian Ocean and Japan. Historical data alone were not sufficient to define possible extreme scenarios, such as the 2004 and 2011 tsunami events [25]. Instead, PTHA offers a systematic way to consider uncertainties associated with tsunami sources, occurrence probability, tsunami generation, wave propagation, and inundation of land areas. It is essential to recognize that PTHA is not a solution, instead, it is a framework to be explicit about what we know and do not know. PTHA also incorporates uncertainties related to tsunami hazard assessments transparently, and sensitivity analysis can be performed to examine the influence of adopted assumptions.

Figure 4 depicts a general computational flow of PTHA, and its main model components are explained in the following. The methodology is implemented in Monte Carlo simulations, thereby numerous stochastic event catalogs are generated. The PTHA model only considers the full-margin rupture pattern of the Cascadia subduction zone because the southern-margin and central-margin rupture patterns generate significantly smaller tsunamis along the Canadian coast due to the directivity of the generated waves [15]. Ignoring the southern-margin and central-margin rupture cases results in the underestimation of tsunami hazard. However, the extent of the underestimation is relatively small due to the above-mentioned radiation characteristics of tsunami waves. The earthquake recurrence is based on a time-dependent renewal process [22] and can incorporate the elapsed time since the last event in defining the probability distribution of the inter-arrival time for the first event in each stochastic event catalog (Section 3.1). The magnitude model also needs to be specified (Section 3.2). For the earthquake source modeling, the adopted approach is based on stochastic rupture models for the Cascadia megathrust events [15]. The consideration of stochastic rupture models is advantageous to incorporate the uncertainty associated with earthquake rupture processes in terms of fault plane geometry, fault plane position, and heterogenous earthquake distribution (Section 3.3.1). The stochastic rupture approach is capable of capturing the uncertainty of the earthquake rupture processes more widely than

the conventional approaches that employ the uniform earthquake slip distribution together with a logic tree model [26,27]. Subsequently, a tsunami propagation model for the Cascadia region is set up to simulate tsunami waves at various nearshore locations at shallow depths (Section 3.3.2). Okada [28] equations and Tanioka and Satake [29] formulae are then used to compute the ground displacements due to a fault rupture, whereas nonlinear shallow water equations are solved using the finite-difference method [30]. In Section 3.4, the computational aspects of integrating different model components are explained.

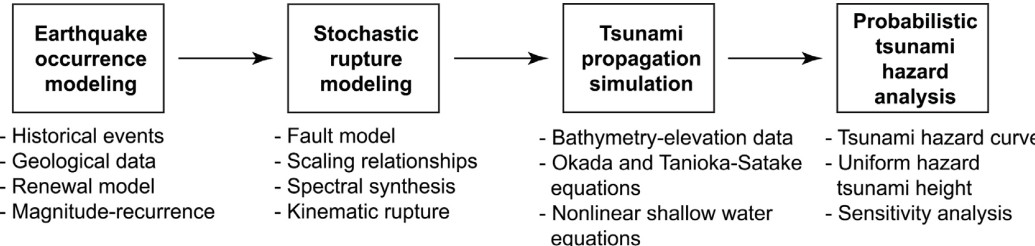

**Figure 4.** Computational steps for probabilistic tsunami hazard analysis.

*3.1. Earthquake Occurrence Model*

3.1.1. One-Component Renewal Model

A renewal process offers a flexible mathematical method for characterizing the earthquake recurrence from major subduction zones. The process is characterized by the inter-arrival time distribution and the elapsed time since the last event. By adopting different inter-arrival time distributions, variable recurrence behavior can be introduced into PTHA [22]. For the simplest case, the exponential distribution can be adopted for modeling the inter-arrival times between successive events, noting that the model parameter is the mean recurrence period. For the case of the exponential distribution, the hazard rate function, which describes the rate of earthquake occurrence for a given time by reflecting the effect of the elapsed time since the last event, becomes constant, representing a memory-less property of the time-independent Poisson process. More common types for the inter-arrival time distribution of major earthquakes include the normal distribution, lognormal distribution, Brownian Passage Time distribution [18], and Weibull distribution [19]. These one-component renewal models typically have two model parameters to define the inter-arrival time distribution function: Mean recurrence period and coefficient of variation. When the coefficient of variation is relatively small (e.g., less than 0.5), the renewal model exhibits quasi-periodic recurrence characteristics of the major events.

To illustrate a standard one-component renewal model, in Figure 5a, the fitted inter-arrival time models for the normal and Weibull distributions are compared with the histogram of the resampled inter-arrival time data for the Cascadia megathrust events (Section 2.3). The Weibull distribution is selected because it is the most preferred one-component renewal model among those tested according to the computed values of the Akaike Information Criterion (*AIC*):

$$AIC = 2N_p - 2lnL \qquad (1)$$

where $N_p$ is the number of model parameters, and *lnL* is the loglikelihood value of the model. The model fitting is carried out using the maximum likelihood method, and for the same data, a model with a smaller *AIC* value is superior. Due to the unsymmetrical characteristics of the resampled inter-arrival time data, the fit of the normal distribution is not particularly good. The Weibull distribution fits better with the resampled inter-arrival time data, but there are still notable misfits at the modal peak (underestimation) and both sides of the peak (overestimation). Although the exponential distribution (i.e., time-independent Poisson process, see the red line in Figure 5a) is simple and popular, the shape of the distribution is not adequate to capture the characteristics of the resampled inter-arrival time data.

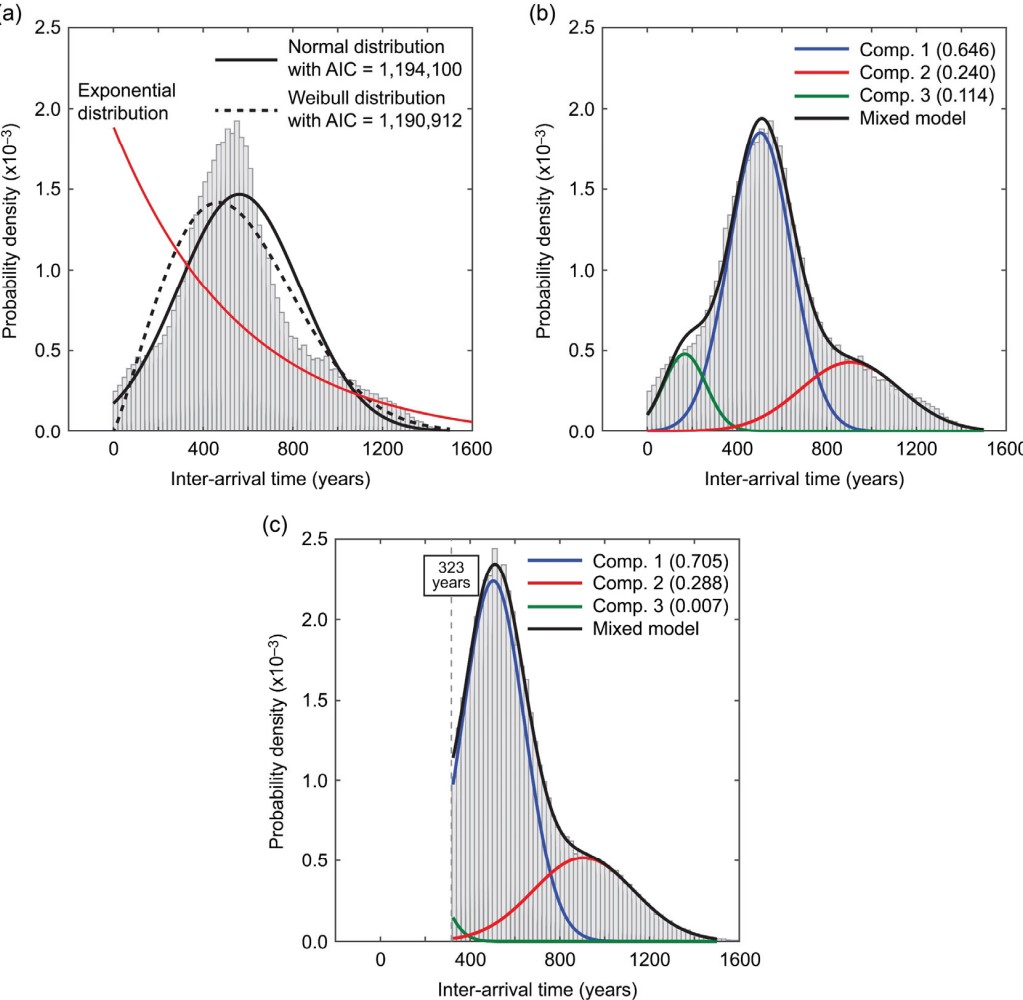

**Figure 5.** (**a**) Comparison of (one-component) renewal models with the normal (Gaussian) and Weibull inter-arrival time distributions. (**b**) Three-component Gaussian mixture model. (**c**) Three-component Gaussian mixture model conditioned on the elapsed time of 323 years. In (**b**,**c**), the proportions of the mixed components are indicated in the brackets.

3.1.2. Gaussian Mixture Model

To improve the fit of the analytical function to the resampled inter-arrival time data, a Gaussian mixture model is considered, comprising multiple Gaussian components. The Gaussian mixture model for $K$ components can be expressed by:

$$f(x) = \sum_{i=1}^{K} \pi_k N(x|\mu_k, \Sigma_k) \tag{2}$$

where $\pi_k$ is the mixing proportion of the $k$-th component and its summation over all $K$ components equals 1. The $k$-th mixing proportion represents the probability of observing data that come from the $k$-th Gaussian component. $N(x|\mu_k, \Sigma_k)$ is the Gaussian density function of the $k$-th component and is given by:

$$N(x|\mu_k, \Sigma_k) = \frac{1}{(2\pi)^{0.5D}|\Sigma_k|^{0.5}} \exp\left(-\frac{1}{2}(x-\mu)^T \Sigma_k^{-1}(x-\mu)\right) \tag{3}$$

For the $D$-dimensional data, $\mu_k$ and $\Sigma_k$, are the mean vector and covariance matrix of the $k$-th component. In this study, $D = 1$ since only a single variable is concerned, and the notation for the covariance $\Sigma_k$ in Equations (2) and (3) can be replaced with $\sigma_k^2$.

The parameters of the Gaussian mixture model can be estimated using the Expectation-Maximization (EM) algorithm [31]. The EM algorithm attempts to maximize the loglikelihood function of the Gaussian mixture model for given data in two steps with iterations. For the specified value of $K$, initial values for component means, covariance matrices, and mixing proportions are generated through a $k$-means++ technique. In the Expectation step, the algorithm computes posterior probabilities of component memberships for each data point. Subsequently, in the Maximization step, with the component membership posterior probabilities, component means, covariance matrices, and mixing proportions are estimated based on the maximum likelihood method. The Expectation-Maximization steps are iterated until the convergence is achieved. Since the success of the EM algorithm depends on the data complexity and initial values and the solution may converge to a local minimum, multiple runs of the EM algorithm are made, and the results with the highest loglikelihood value are adopted as the final estimate.

In this study, the Gaussian mixture model is fitted to the simulated inter-arrival time data by considering $K = 3$. Note that $K = 1$ corresponds to the one-component normal distribution case shown in Figure 5a. The Gaussian mixture model with $K = 3$ is superior to that with $K = 2$ according to the *AIC* values (*AIC* = 1,185,057 versus 1,186,767). The results of the three-component Gaussian mixture model fit are shown in Figure 5b. The three components correspond to clusters with mean recurrence periods of 503, 905, and 117 years and standard deviations of 139, 224, and 95 years. The second component corresponds to the long gaps with a mixing proportion of 0.24, whereas the third component corresponds to the short-term clustering with a mixing proportion of 0.114 (Figure 5b). Overall, the three-component Gaussian mixture model captures the resampled inter-arrival time data well (as also revealed in the *AIC* value).

To apply the developed three-component Gaussian mixture model as the inter-arrival time distribution in time-dependent PTHA, the elapsed time since the last event needs to be taken into account. For this purpose, the mixing proportions of the three components are updated. More specifically, the probability that the inter-arrival time is longer than the elapsed time is calculated for each component, and then multiply this probability by the original mixing proportion of the component. Once the same calculations are performed for all three components, the updated mixing proportions are computed by normalizing these quantities to 1.0. For the elapsed time $T_E$ of 323 years, the updated three-component Gaussian mixture model is shown in Figure 5c. The updated mixing proportions are indicated inside the brackets in the figure legend. At the present time, the relative likelihood of the third component (i.e., short-term clustering) is very small compared to the other two components.

### 3.2. Earthquake Magnitude Model

The earthquake magnitude distribution is important for tsunami hazard assessments. The magnitudes of the Cascadia subduction events are primarily dependent on the rupture patterns and corresponding rupture areas (i.e., segmented versus whole ruptures). In this study, two magnitude models are considered. The first one is the Gutenberg–Richter model with a $b$-value of 1. The second one is based on the characteristic magnitude model with a uniform distribution (for the characteristic portion only). The above-mentioned two magnitude models can be considered as two end-member models for the magnitude distribution. Since the whole rupture scenarios are concerned, the minimum and maximum magnitudes are set to 8.7 and 9.1 for both magnitude models. The occurrence probabilities of the full-margin megathrust Cascadia events are controlled by the earthquake occurrence models (Sections 3.1.1 and 3.1.2). The two magnitude models are shown in Figure 6, noting that the magnitude range is discretized into four bins with 0.1 magnitude width. The characteristic uniform magnitude model assigns larger weights to the $M9.0$–9.1 bin. Hence, the tsunami hazard is expected to be greater when the characteristic uniform magnitude model is adopted instead of the Gutenberg–Richter model.

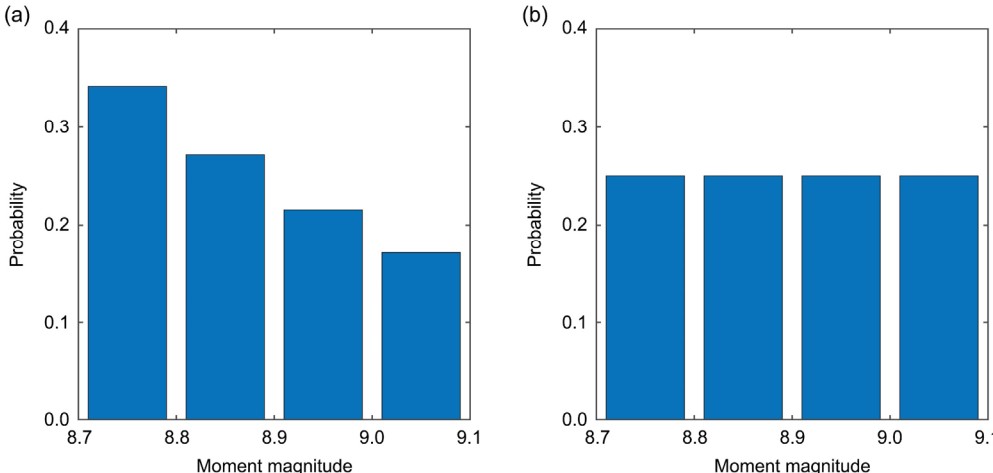

**Figure 6.** (**a**) Conditional magnitude probability distribution based on the Gutenberg–Richter relationship with $b = 1$. (**b**) Conditional magnitude probability distribution based on the characteristic uniform model.

### 3.3. Stochastic Tsunami Simulations

To consider a wide range of possible earthquake rupture scenarios and patterns for the Cascadia megathrust earthquakes and to quantify the uncertainty of tsunami hazard estimates for the Canadian coast, stochastic rupture models for the full-margin megathrust Cascadia events are generated [15]. The number of synthesized stochastic rupture models is 2000, and they have earthquake magnitudes between 8.7 and 9.1. The stochastic tsunami simulations involve a sequence of modeling and numerical analyses: (i) Selection of a fault plane model, (ii) use of statistical scaling relationships for earthquake source parameters, (iii) random earthquake slip generation, (iv) ground displacement estimation, and (v) tsunami inundation simulation. In this section, key features of the stochastic rupture models for the Cascadia megathrust earthquakes are described. Full details of the source models can be found in [15]. Figure 7 illustrates the main computational steps of stochastic tsunami hazard simulations for the Cascadia megathrust earthquakes.

#### 3.3.1. Stochastic Source Models

The fault plane geometry for the Cascadia megathrust earthquakes is based on the Slab2 model [32]. It represents a complex curved feature of the Cascadia megathrust interface with variable strike and dip angles. This fault plane is approximated by a set of 7452 sub-faults, reaching depths of 30 km and each having a size of 5.6 km along the strike and 3.8 km along the dip direction (Figure 7a). Because the stochastic source method primarily works with a rectangular 2D surface, the irregular 3D sub-faults are mapped to an overall 2D fault plane, consisting of $201 \times 51$ sub-fault cells. Since some portions of the 2D sub-faults are not associated with the 3D sub-faults, these unmatched sub-faults are masked when generating 2D stochastic source models. This mapping process is illustrated in Figure 7a.

To synthesize an earthquake slip distribution over the 2D sub-faults, a scenario magnitude is specified within a 0.1 magnitude bin [15], and a magnitude value is sampled from the uniform distribution. For the simulated magnitude value, eight earthquake source parameters, i.e., fault length $L$, fault width $W$, mean slip $D_a$, maximum slip $D_m$, Box-Cox parameter $\lambda_{BC}$, along-strike correlation length $CL_L$, along-dip correlation length $CL_W$, and Hurst number $H$, are generated from the statistical scaling relationships by [33]. These parameters are represented by the multi-variate lognormal distribution with a correlation structure. In this step, several checks of the sampled source parameters are carried out by ensuring that the ratios of correlation lengths to fault dimensions are consistent with empirical ranges of 0.15–0.6 for the strike direction and 0.15–0.45 for the dip direction.

Eventually, the simulated values of $L$, $W$, and $D_a$ must fall within the target magnitude range. Figure 7b shows the simulated fault area (i.e., $L \times W$) and mean slip $D_a$ for 5000 stochastic source models having moment magnitudes between 8.1 and 9.1 [15]. Once a suitable fault geometry is determined, the fault plane is placed randomly within the overall fault plane (i.e., 201-by-51 rectangular matrix).

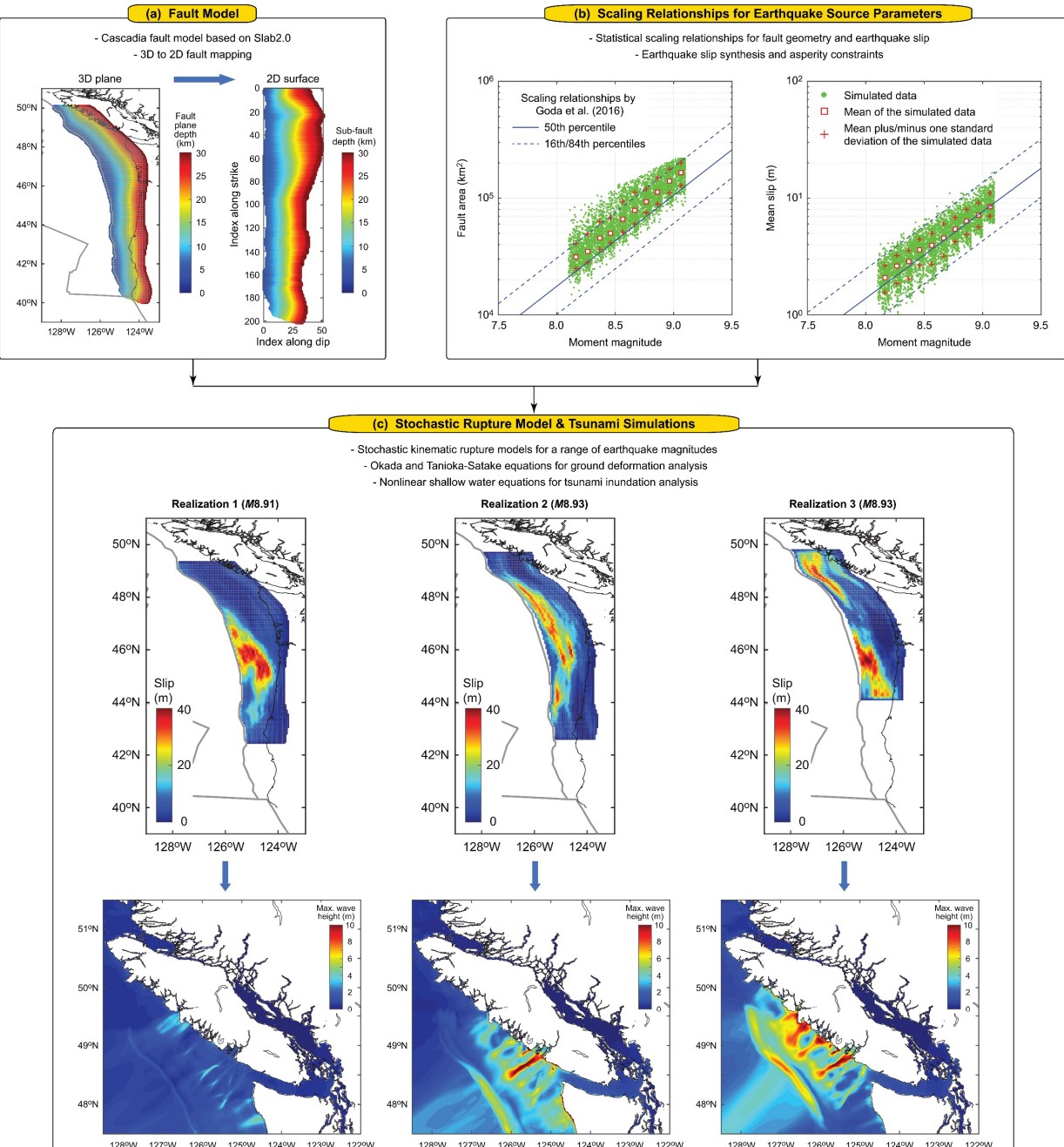

**Figure 7.** Stochastic tsunami simulations for the Cascadia megathrust earthquakes: (**a**) fault model, (**b**) scaling relationships for earthquake source parameters, and (**c**) stochastic rupture model and tsunami simulations.

For a given fault plane geometry, a heterogeneous earthquake slip distribution is synthesized. A candidate slip distribution is first simulated from an anisotropic 2D von Kármán wavenumber spectrum with its amplitude spectrum being parametrized by the three parameters $CL_L$, $CL_W$, and $H$ and its phase being randomly distributed between 0 and

$2\pi$ [34]. The simulated slip distribution is modified via Box–Cox power transformation to achieve a desirable right-skewed feature of the marginal slip distribution [33]. At this stage, the edge tapers are implemented on the northern, southern, and eastern boundaries of the stochastic earthquake source models to avoid abrupt changes in the earthquake slip values near the along-dip edges of the fault model. Subsequently, slip values of the sub-faults that are within the masked sub-faults are set to zero, and all eligible slip values are scaled to match the mean slip. To ensure that the simulated earthquake slip distribution has realistic characteristics for the target Cascadia megathrust earthquakes, several constraints on the simulated slip distribution are implemented. For instance, an asperity zone (i.e., earthquake slip concentration within the specified area of the fault plane) is defined by reflecting different rupture patterns identified in the paleo-seismic data [6]. Major asperities are also constrained to occur in the shallow part of the subduction interface to broadly coincide with the outer wedge of the accretionary prism. When the candidate slip distribution does not meet all criteria, this realization is discarded, and another earthquake rupture model is generated. This process is repeated until an acceptable model is obtained.

Finally, by repeating the above-mentioned procedure for earthquake rupture modeling 500 times for each of the ten magnitude bins between $M$8.1 and $M$9.1, a set of 5000 earthquake rupture models was generated [15]. Among these stochastic rupture models, 2000 models with earthquake magnitudes between $M$8.7 and $M$9.1 are used in this study. Three realizations of the $M$8.9–9.0 scenario are shown in Figure 7c. Different stochastic rupture models can exhibit variations in fault plane geometry, position within the overall Cascadia megathrust fault plane, and earthquake slip distribution.

### 3.3.2. Tsunami Propagation Simulation

To effectively perform numerous tsunami simulations for different spatial domains and at specified spatial resolutions, bathymetry and elevation data are merged to develop nesting grid datasets. Deep water bathymetry data are collected from the GEBCO (General Bathymetric Chart of the Oceans) dataset ($\approx$450-m resolution), while shallow water bathymetry data around Vancouver Island are obtained from the CHS (Canadian Hydrographic Service) dataset (10-m resolution). For land areas, the CDEM (Canadian Digital Elevation Model) data ($\approx$20-m resolution) are used. The merged bathymetry-elevation data are then arranged into nested grid systems (810-m and 270-m). The vertical reference datum is at the mean sea level.

For a given earthquake rupture model, a vertical dislocation profile of seawater due to an earthquake rupture is computed using Okada [28] equations. To account for the effects of horizontal movements of seafloor slopes on the vertical dislocation of seawater, a method proposed by [29] is implemented, which is important for the ground deformation of the outer wedge that has greater slope angles than other parts of the bathymetry in the Cascadia subduction zone. To alleviate the abrupt changes of the vertical dislocation of seawater, a spatial smoothing filter of 9-by-9 cells (810-m grids) is employed. The earthquake rupture model is implemented as a kinematic source with variable rupture propagation velocity and rise time. The rise time prediction model by [35] is adopted in this study. Moreover, the effects of coseismic ground deformation are considered by adjusting the elevation data prior to the tsunami simulation run.

The tsunami modeling is carried out using a well-tested TUNAMI code [30] that solves the nonlinear shallow water equations using a leap-frog staggered-grid finite difference scheme and is capable of generating offshore tsunami propagation and onshore run-up. For all computational cells, the bottom friction and surface roughness are represented by a uniform value of Manning's coefficient equal to 0.025 m$^{-1/3}$ s, which is often used for agricultural land and ocean/water [36]. The run-up calculation is based on a moving boundary approach [36], where a dry/wet condition of a computational cell is determined based on total water depth relative to its elevation. Illustrations of three regional tsunami simulations of the $M$8.9–9.0 scenario are shown in Figure 7c, exhibiting the significant effects of the earthquake rupture characteristics on the generated tsunami waves. Each

tsunami simulation is performed for a 3-h duration, which is sufficient to model the most critical phase of tsunami waves for the Cascadia tsunami scenarios. The wave recording locations are set up along the Vancouver Island coast (Figure 1b). The locations have water depths of approximately 10 m. For these nearshore locations, the grid resolution of 270 m is suitable, resulting in the required time-stepping interval of 1.0 s to satisfy the Courant–Friedrichs–Lewy condition.

### 3.4. Computational Procedure

PTHA is carried out based on Monte Carlo simulations, and this procedure is graphically shown in Figure 8. The three main components of PTHA are the earthquake occurrence model (Section 3.1), the magnitude model (Section 3.2), and the stochastic tsunami simulations (Section 3.3). In the earthquake occurrence model (Figure 8a), simulations of the first event and the subsequent events need to be distinguished to account for the effects of the elapsed time since the last major event $T_E$. In the magnitude model (Figure 8b), different magnitude distributions can be considered. In the stochastic tsunami simulations (Figure 8c), the conditional probability distributions of the maximum tsunami wave amplitude at a site of interest can be derived by analyzing the tsunami simulation results for all stochastic rupture models corresponding to the discretized earthquake magnitude bin. The maximum tsunami amplitude is defined as the peak wave height above the baseline vertical datum (i.e., mean sea level).

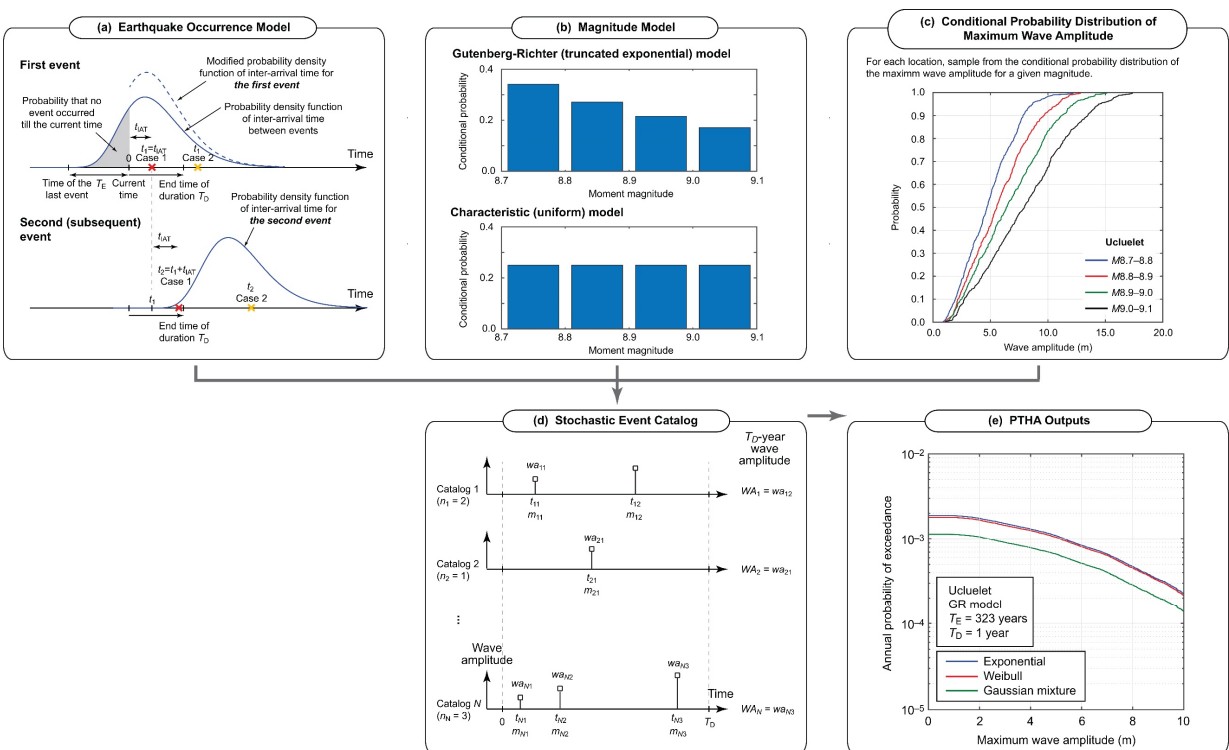

**Figure 8.** Computational procedures for probabilistic tsunami hazard analysis: (**a**) Earthquake occurrence model, (**b**) magnitude model, (**c**) conditional probability distributions of the maximum wave amplitude for different magnitude ranges, (**d**) stochastic event catalogs together with event times, earthquake magnitudes, and maximum wave amplitudes, and (**e**) PTHA outputs, such as tsunami hazard curves at a location of interest (e.g., Ucluelet) with different inter-arrival time distributions.

Through Monte Carlo simulations, stochastic event catalogs, each of which has a duration of $T_D$ years, are generated. For instance, in the $i$-th catalog, $n_i$ events may occur (note: For a relatively short duration of $T_D$, there are many stochastic catalogs that do not have any tsunami events). For each event, a moment magnitude value is sampled from

the considered magnitude model, and then the maximum tsunami amplitude at the site of interest can be sampled from the corresponding conditional probability distribution of the maximum tsunami wave amplitude (Figure 8d). Subsequently, by taking the maximum tsunami wave amplitude over $T_D$ years, the $T_D$-year maxima of the tsunami hazard measure can be obtained. By analyzing the $T_D$-year maxima data from $N$ stochastic catalogs, the $T_D$-year probability of exceedance can be assigned to each of the $T_D$-year maxima data. Eventually, the pairs of the $T_D$-year maxima data and the $T_D$-year probability of exceedance can be displayed as a tsunami hazard curve (Figure 8e). By repeating the same analysis but considering different model components and parameters, multiple tsunami hazard curves that represent alternative assumptions can be obtained. This is a form of sensitivity analysis. In the above-mentioned procedure, a logic tree model can be integrated into the simulation-based PTHA procedure (e.g., by considering equal weights for the two magnitude models) to incorporate the effects of epistemic uncertain models and parameters into the tsunami hazard assessments.

## 4. Regional Tsunami Hazard Assessment for Vancouver Island Due to the Cascadia Subduction Earthquakes

### 4.1. Analysis Set-Up

This section presents the results of regional PTHA for offshore locations along the Vancouver Island coast. Six locations are selected to show site-specific PTHA results (Section 4.2). In addition, to show uniform tsunami hazard curves at a regional scale, 109 locations are focused upon, whose depths are approximately 10 m between 7 and 13 m (Section 4.3). The above-mentioned locations are indicated in Figure 1b.

The PTHA is conducted on a site-specific basis. As the base case, the elapsed time since the last event is set to $T_E$ = 323 years (present situation), and the duration of the assessment is set to $T_D$ = 1 year. To evaluate the influence of different values of $T_E$ and $T_D$, they are varied to: $T_E$ = 353 years and 383 years and $T_D$ = 50 years. For the inter-arrival time distribution, three models are considered: Exponential distribution (i.e., time-independent Poisson process), Weibull distribution (i.e., best one-component renewal model), and three-component Gaussian mixture model (see Figure 5). For the magnitude model, both the Gutenberg–Richter model (i.e., truncated exponential distribution) and the characteristic uniform model are considered (Figure 6). The uncertainty of the earthquake rupture process is reflected in the conditional probability distribution of the offshore maximum wave amplitude for each magnitude bin (Figures 7 and 8c). The PTHA results are presented in the form of a site-specific tsunami hazard curve (i.e., plot of the maximum tsunami amplitude against the $T_D$-year probability of occurrence) and a uniform tsunami hazard curve (i.e., plot of the maximum tsunami amplitudes for multiple offshore locations at a specified $T_D$-year probability of occurrence level). The simulation number is set to 50 million times when $T_D$ = 1 year and 1 million times when $T_D$ = 50 years (i.e., the total number of years is kept the same).

### 4.2. Site-Specific Tsunami Hazard Curves

Figure 9 shows tsunami hazard curves for six locations indicated in Figure 1b by considering three inter-arrival time distributions with $T_E$ = 323 years and $T_D$ = 1 year. The magnitude model is the Gutenberg–Richter relationship (Figure 6a). Among the six locations, the tsunami hazards can be ordered from high to low as Ucluelet, Bamfield, Tofino, Yuquot, Port Renfrew, and Metchosin, noting that tsunami hazard levels for Bamfield and Tofino are similar. Both Ucluelet and Bamfield are strongly affected by offshore underwater topography that creates more concentrated tsunami waves along submarine canyons (e.g., Barkley Canyon) (note: The selected location for Bamfield is slightly closer to the land and is less exposed to the Pacific Ocean, see Figure 1b). This can be seen in Figure 7c. Tofino also has a high tsunami hazard due to its direct exposure to the Pacific Ocean. Although Yuquot is exposed to the Pacific Ocean, its tsunami hazard level is less than Tofino because it is located at the northern end of the Cascadia subduction zone. The tsunami hazard

levels of Port Renfrew and Metchosin are less than the preceding four locations because they are located along the Juan de Fuca Strait. In other words, the relative tsunami hazards can be understood from the physical oceanographic conditions of the locations with respect to the Cascadia subduction zone.

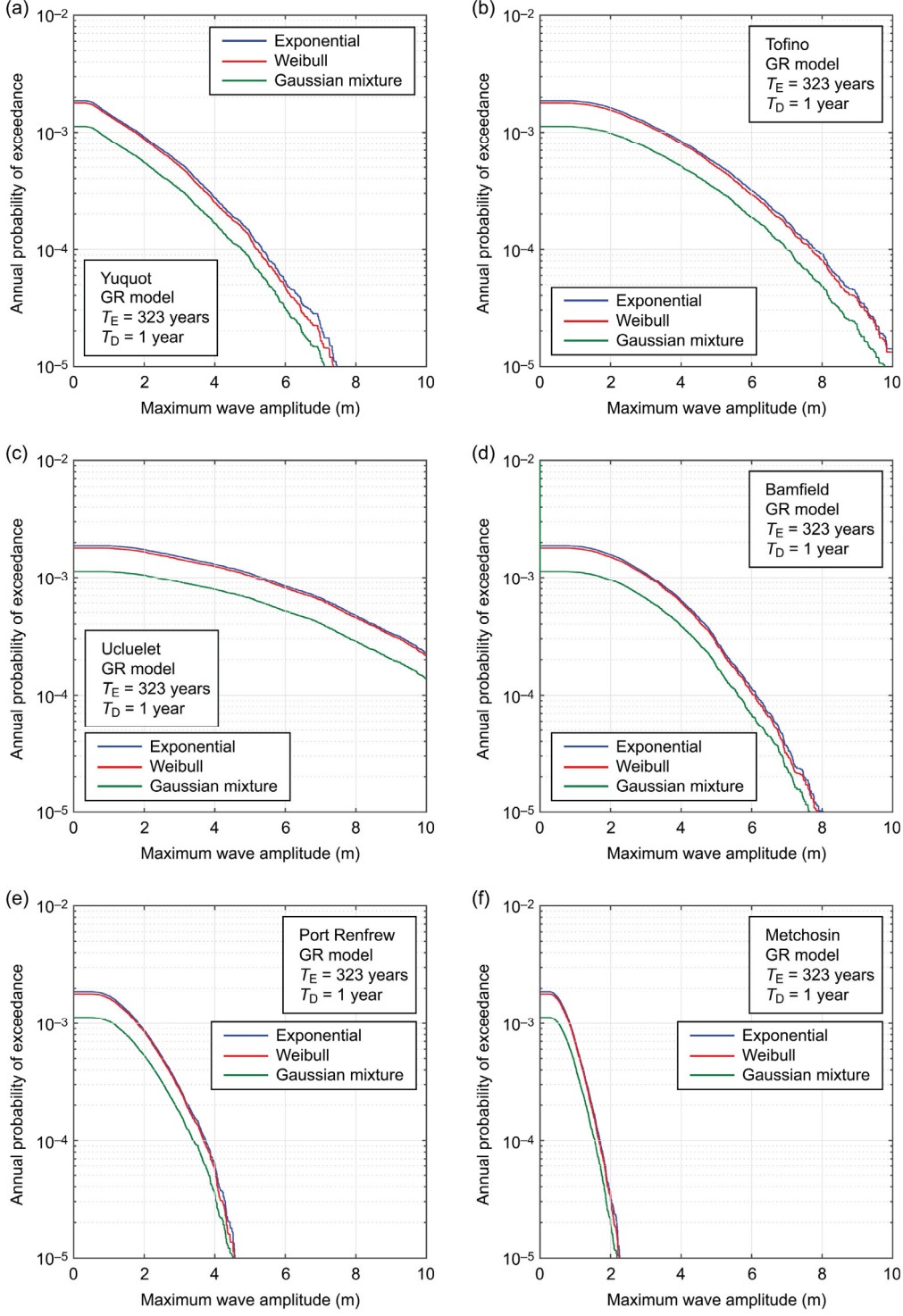

**Figure 9.** Tsunami hazard curves by considering three inter-arrival time distributions with $T_E = 323$ years and $T_D = 1$ year: (**a**) Yuquot, (**b**) Tofino, (**c**) Ucluelet, (**d**) Bamfield, (**e**) Port Renfrew, and (**f**) Metchosin. The magnitude model is based on the Gutenberg–Richter distribution.

For a given location, the relative positions of the three tsunami hazard curves based on the exponential, Weibull, and Gaussian mixture distributions are consistent. For $T_E$ = 323 years, the cases with the Weibull distribution and the exponential distribution produce higher tsunami hazards than the case with the Gaussian mixture model (note: The first two cases are similar, but the exponential distribution produces slightly larger tsunami hazard estimates). This can be understood by inspecting the probability distributions of the simulated inter-arrival time for the three distributions. These are shown in Figure 10a for the Weibull distribution and Figure 10c for the Gaussian mixture distribution, respectively. Note that the inter-arrival time 0 corresponds to the chosen value of $T_E$ (=323 years). The exponential probability distribution (which does not depend on $T_E$) is also shown in the figure panel and can be used as a reference for the time-independent tsunami hazard case. When the inter-arrival time is less than one year (i.e., $T_D$ = 1 year), the probability density value for the Weibull distribution is slightly larger than the exponential distribution (but the values are close), whereas the probability density value for the Gaussian mixture distribution is noticeably smaller than the exponential distribution.

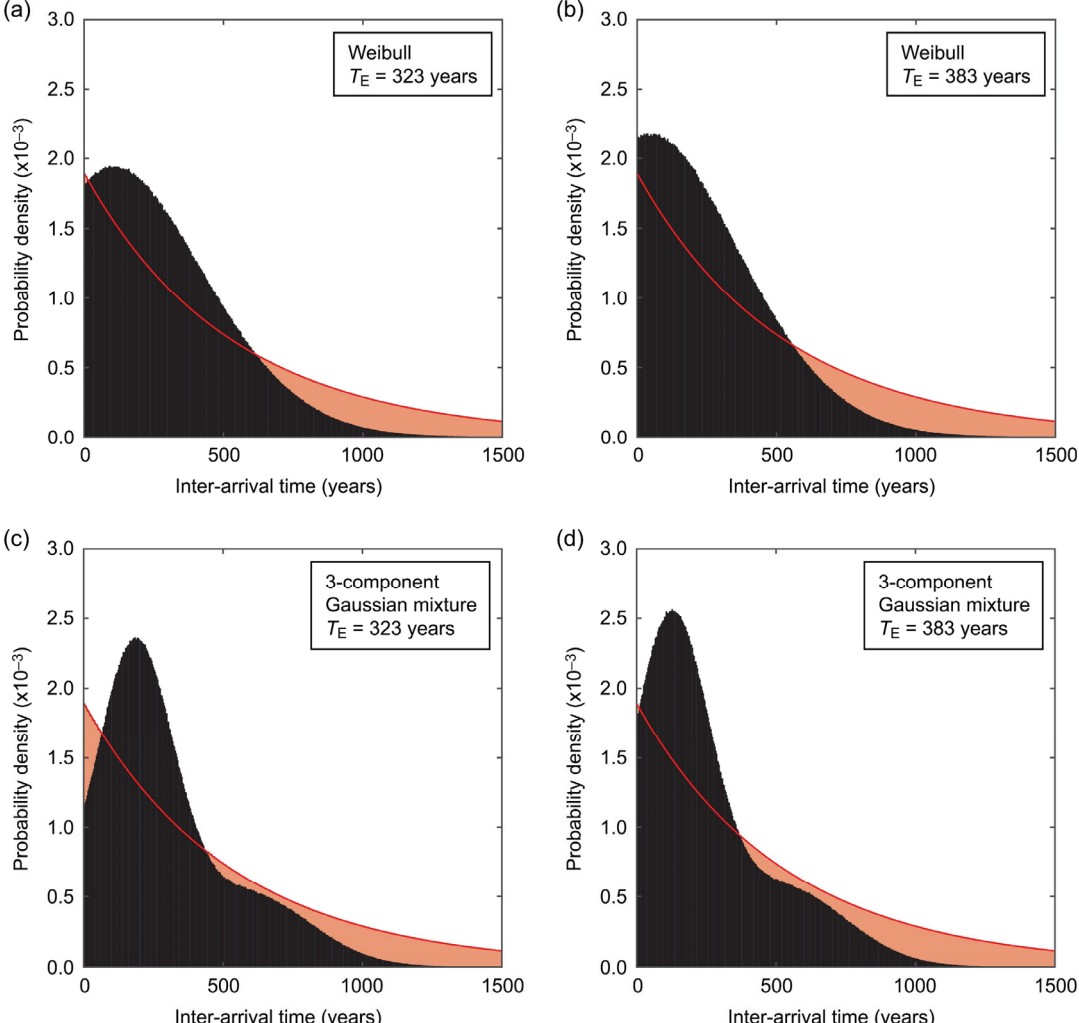

**Figure 10.** Updated inter-arrival time distributions: (**a**) Weibull distribution with $T_E$ = 323 years, (**b**) Weibull distribution with $T_E$ = 383 years, (**c**) Three-component Gaussian mixture distribution with $T_E$ = 323 years, and (**d**) Three-component Gaussian mixture distribution with $T_E$ = 383 years. The red curve (i.e., time-independent Poisson process) is the same in every figure panel.

To investigate the effects of $T_E$ and $T_D$ as well as magnitude distributions on the tsunami hazard curves, Tofino is focused upon. The observations made below are also

applicable to the five other locations. Figure 11 shows tsunami hazard curves for Tofino by considering three inter-arrival time distributions but with different values of $T_E$ and $T_D$ and different magnitude distributions. The comparison of Figure 11a,b indicates that the consideration of the characteristic uniform distribution results in higher tsunami hazard curves, as expected. This is because the uniform model assigns higher relative weights to large-magnitude events, and when the magnitudes are greater, the tsunami wave amplitudes tend to be greater (Figure 8c).

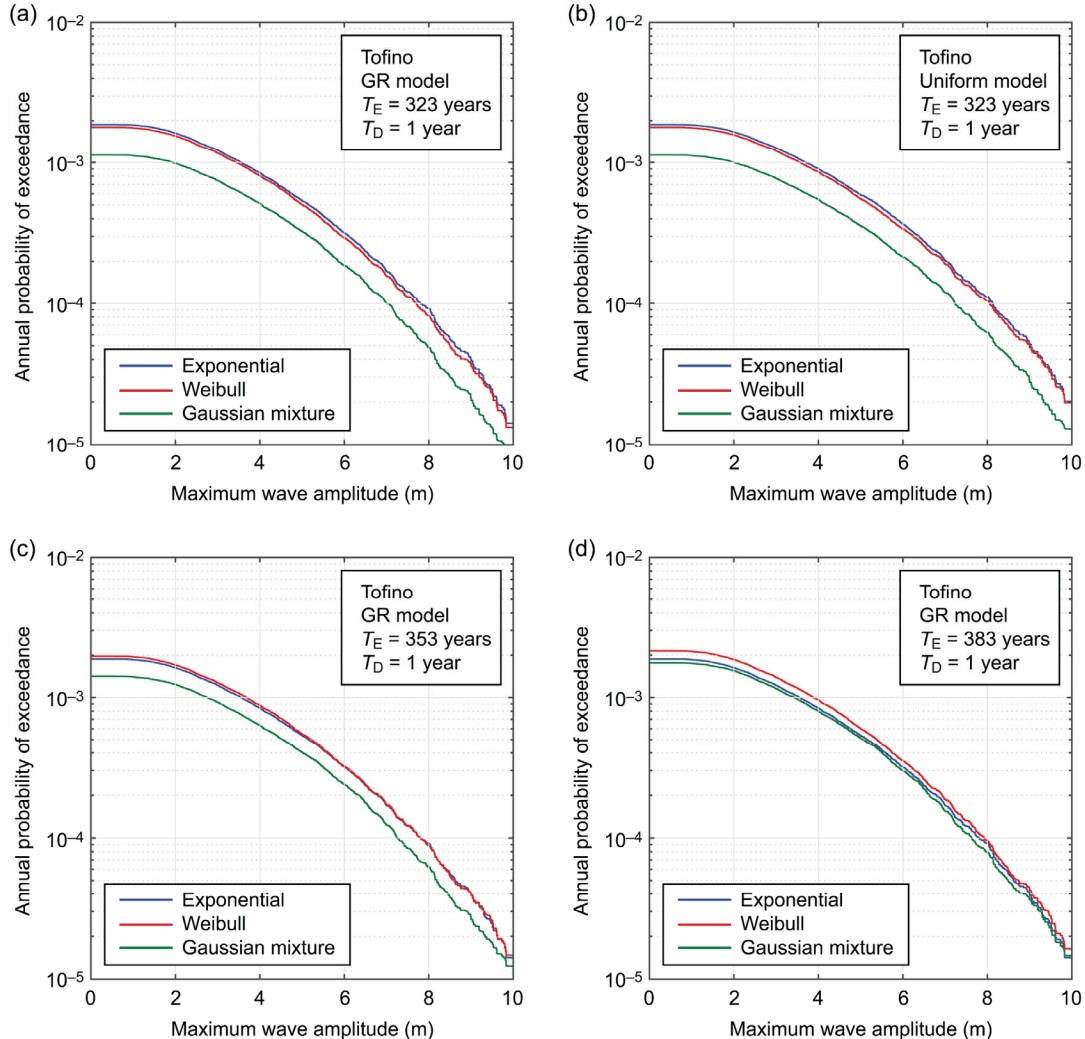

**Figure 11.** Tsunami hazard curves for Tofino by considering three inter-arrival time distributions: (**a**) $T_E$ = 323 years, $T_D$ = 1 year and Gutenberg–Richter model, (**b**) $T_E$ = 323 years, $T_D$ = 1 year and uniform model, (**c**) $T_E$ = 353 years, $T_D$ = 1 year and Gutenberg–Richter model, and (**d**) $T_E$ = 383 years, $T_D$ = 1 year and Gutenberg–Richter model. The blue curve (i.e., time-independent Poisson process) is the same in every figure panel (for the same magnitude model).

The comparison of Figure 11a,c,d demonstrates the effects of $T_E$ for a fixed value of $T_D$ = 1 year. With the increase of $T_E$, the updated inter-arrival time distributions of the Weibull and Gaussian mixture models become higher for the inter-arrival time of less than one year. The inter-arrival time distributions for the Weibull and Gaussian mixture models for $T_E$ = 383 years are shown in Figure 10b,d, respectively. The same increasing trends can be observed in the corresponding tsunami hazard curves shown in Figure 11c,d. When $T_E$ = 353 years, the case with the Weibull distribution produces a larger tsunami hazard curve (by a slight margin) than the exponential distribution, while the tsunami hazard curve for the Gaussian mixture model is still below the exponential distribution

(Figure 11c). Considering a situation further into the future ($T_E$ = 383 years), the case with the Weibull distribution is noticeably larger than the exponential distribution, whereas the case with the Gaussian mixture distribution becomes close (but slightly less than) the exponential distribution.

Overall, considering that the Gaussian mixture distribution is the most consistent with the underlying inter-arrival times of the Cascadia megathrust events, the overestimation of the tsunami hazard at the present time when the simpler one-component renewal model or the Poisson model can be as large as 1 m at a given annual probability of exceedance level for the case of Tofino (e.g., taking the horizontal differences of the tsunami hazard curves).

Moreover, the effects of $T_D$ on the tsunami hazard curves for Tofino are investigated in Figure 12, with respect to the varied $T_E$ values. More specifically, Figure 11a,c,d corresponds to Figure 12a–c, respectively. When $T_D$ is longer than one year, the area under the inter-arrival time distribution should be compared for different inter-arrival time distributions. Inspecting Figure 11, when $T_E$ = 323 years, the exponential distribution still produces the larger probability values (thus producing larger tsunami hazards). However, when $T_E$ = 383 years, the areas under the time-dependent earthquake occurrence models become greater than the area under the time-independent earthquake occurrence model. Therefore, in Figure 12c, both time-dependent tsunami hazard curves exceed the time-independent tsunami hazard curve, which is different from the results shown in Figure 11d.

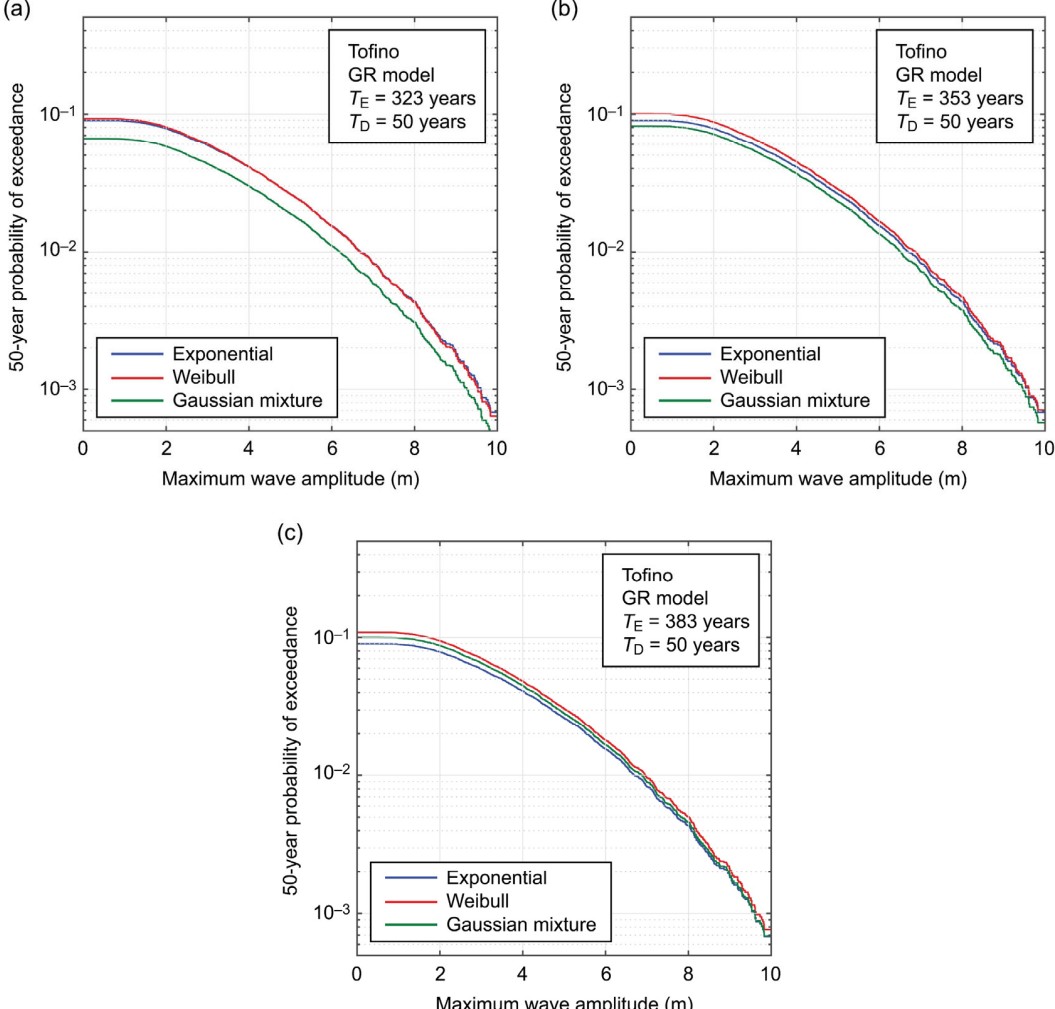

**Figure 12.** Tsunami hazard curves for Tofino by considering three inter-arrival time distributions with $T_D$ = 50 years: (**a**) $T_E$ = 323 years, (**b**) $T_E$ = 353 years, and (**c**) $T_E$ = 383 years. The magnitude model is based on the Gutenberg–Richter distribution.

### 4.3. Regional Uniform Tsunami Hazard Curves

To investigate how the probabilistic tsunami estimates vary along the Vancouver Island coast, uniform tsunami hazard curves are derived from the PTHA results. The site locations for these uniform tsunami hazard curves are shown in Figure 1b. For the assessments, four return period levels, i.e., $T_R$ = 1000, 2500, 5000, and 10,000 years, are considered. Figure 13 shows such uniform tsunami hazard curves by considering three inter-arrival time distributions with $T_E$ = 323 years and $T_D$ = 1 year. Consistently across the four return period levels, the maximum tsunami amplitudes are high in the latitude range between 48.6° and 49.6°. This range approximately corresponds to the coastal line from the south of Yuquot to the north of Port Renfrew. There are several peaks in the uniform tsunami hazard curves, where the focusing effects are induced by the local submarine valleys (Figure 7c). As expected, the tsunami hazard estimates increase with the return period, and their maximum values can exceed 15 m for extreme situations. This indicates that the shoaling effects can further amplify these offshore tsunami wave amplitudes. The detailed run-up and inundation effects must be quantified using high-resolution digital elevation data, which is beyond the scope of this study.

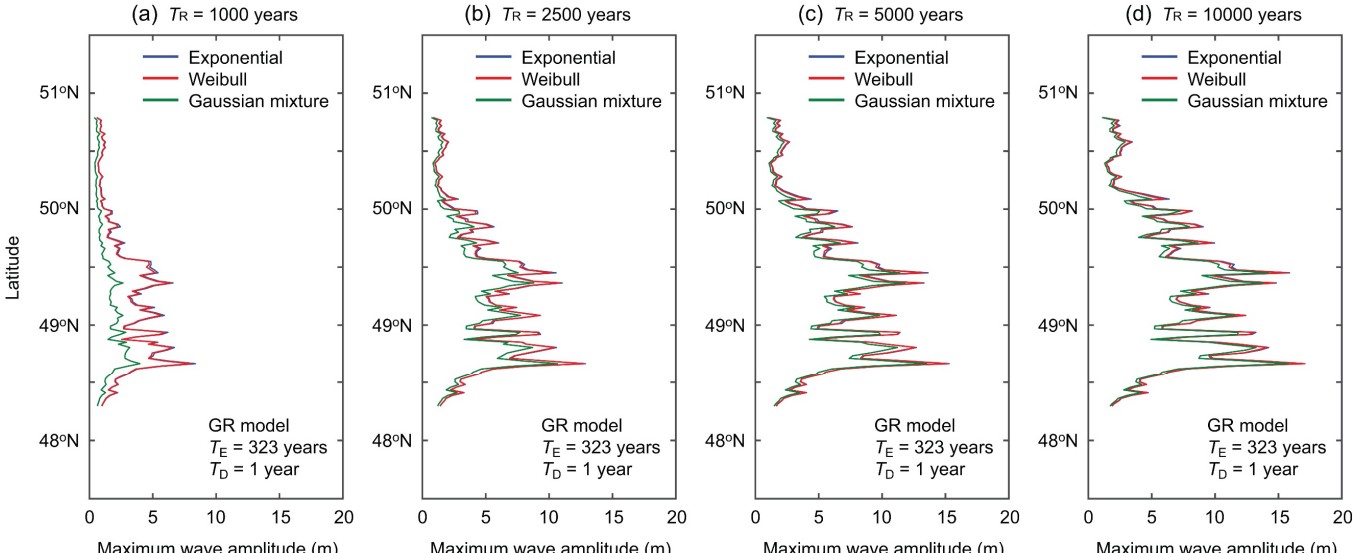

**Figure 13.** Uniform tsunami hazard curves at 109 offshore locations by considering three inter-arrival time distributions with $T_E$ = 323 years and $T_D$ = 1 year: (**a**) $T_R$ = 1000 years, (**b**) $T_R$ = 2500 years, (**c**) $T_R$ = 5000 years, and (**d**) $T_R$ = 10,000 years.

Moreover, to show similar results but for different values of $T_E$ and $T_D$, uniform tsunami hazard curves based on $T_E$ = 383 years and $T_D$ = 50 years are shown in Figure 14. Since the equivalent return period values are considered in Figures 13 and 14, the results are similar. A noticeable difference is that when a longer elapsed time since the last event (i.e., a remote future situation) and the longer time horizon for the tsunami hazard assessment, the time-dependent earthquake occurrence models produce greater tsunami hazard estimates compared with the time-independent earthquake occurrence model. Consequently, for the considered case, the differences of the three inter-arrival time distributions become smaller compared with the base (present) case shown in Figure 13. Overall, uniform tsunami hazard curves at nearshore sites provide useful information on relative tsunami hazards along the coastal line and can be used to identify the hot spots that are affected by the local topography and submarine channel systems.

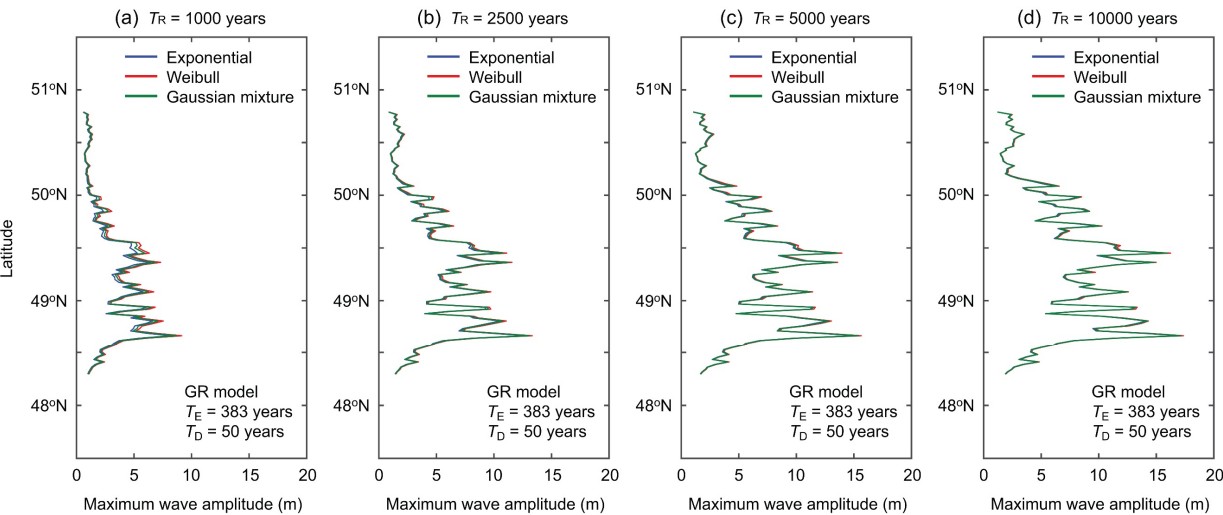

**Figure 14.** Uniform tsunami hazard curves at 109 offshore locations by considering three inter-arrival time distributions with $T_E$ = 383 years and $T_D$ = 50 years: (**a**) $T_R$ = 1000 years, (**b**) $T_R$ = 2500 years, (**c**) $T_R$ = 5000 years, and (**d**) $T_R$ = 10,000 years.

## 5. Conclusions

This study developed the probabilistic tsunami hazard model using stochastic ruptures of the Cascadia megathrust earthquakes. This work is the first of its kind for the locations along the Vancouver Island coast and constitutes the essential step to conduct more detailed probabilistic tsunami inundation hazard and tsunami risk assessments for coastal communities on Vancouver Island. The adopted stochastic rupture modeling approach enabled more comprehensive considerations of possible Cascadia megathrust rupture scenarios in terms of fault geometry, position, and earthquake slip distribution within the overall subduction zone. The developed model also facilitated the consideration of time-dependent earthquake occurrence models by considering suitable inter-arrival time distributions for the Cascadia megathrust events. In particular, the use of the three-component Gaussian mixture distribution was suggested based on the resampled Cascadia subduction age data that reflect underlying uncertainties of the geological data.

The numerical examples of the developed probabilistic tsunami hazard model to the six specific sites and 109 offshore locations at approximately 10 m depths indicate that the consideration of different inter-arrival time distributions can result in noticeable differences in terms of site-specific tsunami hazard curves and uniform tsunami hazard curves at different return period levels. At present, the use of the one-component renewal model tends to overestimate the tsunami hazard values compared to the three-component Gaussian mixture model. With the increase in the elapsed time since the last event and the duration of tsunami hazard assessment (within the parameter variations considered in this study), the differences tend to be smaller. Inspecting the regional variability of the tsunami hazards, specific parts of the Vancouver Island coast are likely to experience higher tsunami hazards due to the directed tsunami waves from the main subduction zone and due to the local underwater topography (i.e., wave focusing through local submarine valleys).

There are several limitations to the presented approach and the obtained results. Firstly, the tsunami sources are limited to megathrust events, while the moderate earthquakes rupturing the southern and central segments of the Cascadia subduction zone (but not extending to the northern segment off Vancouver Island) were not included in the assessments. Note that the exclusion of the southern and central rupture scenarios can be justified on the basis of tsunami radiation patterns for these scenarios. Secondly, the tsunami hazard simulations were performed at a regional scale using the 270-m grids. This grid resolution is too crude to evaluate the tsunami run-up and inundation on land. Although brute-force computations of high-resolution tsunami simulations are possible,

from practical viewpoints, more efficient stochastic tsunami simulation approaches are required to reduce the computational burdens in the future. These improvements will open a new avenue to perform fully probabilistic tsunami inundation risk assessments for coastal communities. Thirdly, although this study presented sensitivity analysis results due to different earthquake occurrence and magnitude models, a fully developed logic tree model will be necessary to capture the effects of epistemic uncertainty on the final tsunami hazard estimates.

**Funding:** The work is funded by the Canada Research Chair program (950-232015) and the NSERC Discovery Grant (RGPIN-2019-05898). The APC was funded by the same funding programs.

**Data Availability Statement:** The results can be requested from the author.

**Conflicts of Interest:** The author declares no conflict of interest.

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
