# Peer review of "Probabilistic Tsunami Hazard Analysis for Vancouver Island Coast Using Stochastic Rupture Models for the Cascadia Subduction Earthquakes"

_2624-795X, doi:10.3390/geohazards4030013_

Round 1

Reviewer 1 Report

The manuscript titled “Probabilistic Tsunami Hazard Analysis for Vancouver Island Coast Using Stochastic Rupture Models for the Cascadia Subduction Earthquakes” by Dr. Goda presents a new probabilistic tsunami hazard model for Vancouver Island from the Cascadia megathrust subduction events. I believe it is a good contribution to the literature. The manuscript can be accepted after a minor revision. Below are my comments.

Line 156: How to understand and confirm the spatial correlation between the northern and southern portions of the subduction zone?

Line 208: How can we interpret the right-skewed nature of the inter-arrival time data and what is the underlying physical significance?

Line 223: Offshore locations typically have less significance in tsunami hazard analysis, thus focusing on "nearshore" is sufficient.

Line 239: In the PTHA model, the author focuses only on the full-margin rupture pattern of the Cascadia subduction zone. Could this approach potentially result in an overestimation of the potential tsunami risk?

Line 352: What is the main purpose of using a characteristic magnitude model? What are its limitations compared to the model based on the Gutenberg-Richter (GR) relationship?

Line 448: What is the rise time of the earthquake? Do you consider instantaneous rupture? In addition, do you consider the effects of rupture direction?

Line 507: Where is the “location of interest”? Please specify it on the map. Actually, I recommend that authors should plot the tsunami hazard curves at more locations instead of just focusing on a map.

Author Response

Please see the attached PDF file.

Reviewer 2 Report

This manuscript presents the first probabilistic tsunami hazard analysis model for key locations on Vancouver Island using stochastic rupture scenarios of the Cascadia megathrust subduction zone. The study found that the use of a 3-component Gaussian component mixture distribution model provides an improved representation of tsunami wave heights at nearshore and offshore locations, compared with the use of a 1-component renewal model which overestimates the hazard. The author highlights the uncertainties in the study, with particular note on the coarse resolution which the simulations were performed which limit an evaluation of tsunami runup and inundation corresponding to the scenario models.

The manuscript is well written with the figures and tables well-presented. The topic of PTHA, in general, is an area of ongoing and growing interest within the seismic and tsunami community and is highly relevant in coastal management and hazard adaptation/mitigation considerations.

Apart from a few very minor queries which I have listed below that centre on clarifying some of the context and also figure presentations, I am pleased to recommend the paper for publication once the author has considered the queries listed below.

1. In Section 2, I think it would be useful to add a section on historical tsunamis, or perhaps absence of, and some background on any paleo-tsunami investigations which have been carried out at, or adjacent to, the study site. I feel this will also help to set the context especially in relation to major historical ruptures which have occurred along the Cascadia megathrust zone. For example, the January 1700 event mentioned in line 37 could be expanded on in section 2 (as an example).

2. Lines 254-255 indicate that the Okada equations and Tanioka-Satake formulae were used to compute ground displacements, but the Tanioka-Satake formulae is not indicated in Figure 3 and in Figure 6.

3. Lines 689-695 – the author raises an important point here, and it might be useful to specify “inundation” between “tsunami” and “risk” in line 695. That is, the use of onshore inundation/runup is becoming increasingly important for use in quantitative risk modelling of asset impacts, damages and losses.

Author Response

Please see the uploaded PDF file.
